# Poisoning for Debiasing: Fair Recognition via Eliminating Bias Uncovered in Data Poisoning

### Yi Zhang
Huawei Cloud
Hangzhou, China
zhangyi481@huawei.com

### Zhefeng Wang*
Huawei Cloud
Hangzhou, China
wangzhefeng@huawei.com

### Rui Hu
Beijing Jiaotong University
Beijing, China
rui.hu@bjtu.edu.cn

### Xinyu Duan
Huawei Cloud
Hangzhou, China
duanxinyu@huawei.com

### Yi Zheng
Huawei Cloud
Hangzhou, China
zhengyi29@huawei.com

### Baoxing Huai
Huawei Cloud
Hangzhou, China
huaibaoxing@huawei.com

### Jiarun Han
Beijing Jiaotong University
Beijing, China
23111132@bjtu.edu.cn

### Jitao Sang*
Beijing Jiaotong University
Beijing, China
Peng Cheng Lab, Shenzhen, China
jtsang@bjtu.edu.cn

## Abstract

Neural networks often tend to rely on bias features that have strong but spurious correlations with the target labels for decision-making, leading to poor performance on data that does not adhere to these correlations. Early debiasing methods typically construct an unbiased optimization objective based on the labels of bias features. Recent work assumes that bias label is unavailable and usually trains two models: a biased model to deliberately learn bias features for exposing data bias, and a target model to eliminate bias captured by the bias model. In this paper, we first reveal that previous biased models fit target labels, which resulted in failing to expose data bias. To tackle this issue, we propose *poisoner*, which utilizes data poisoning to embed the biases learned by biased models into the poisoned training data, thereby encouraging the models to learn more biases. Specifically, we couple data poisoning and model training to continuously prompt the biased model to learn more bias. By utilizing the biased model, we can identify samples in the data that contradict these biased correlations. Subsequently, we amplify the influence of these samples in the training of the target model to prevent the model from learning such biased correlations. Experiments show the superior debiasing performance of our method.

## CCS Concepts

• **Social and professional topics** → **Computing / technology policy**; • **Computing methodologies** → **Machine learning**.

---

*Corresponding authors

## Keywords

Fairness in Machine Learning; Fair Recognition; Data Poisoning

**ACM Reference Format:**
Yi Zhang, Zhefeng Wang, Rui Hu, Xinyu Duan, Yi Zheng, Baoxing Huai, Jiarun Han, and Jitao Sang. 2024. Poisoning for Debiasing: Fair Recognition via Eliminating Bias Uncovered in Data Poisoning. In *Proceedings of the 32nd ACM International Conference on Multimedia (MM '24), October 28-November 1, 2024, Melbourne, VIC, Australia.* ACM, New York, NY, USA, 9 pages. https://doi.org/10.1145/3664647.3681524

## 1 Introduction

Despite the significant advancements in neural networks, a persistent challenge remains: neural networks often learn biased correlations between peripheral features and labels, deviating from human intentions [1, 17, 28]. This issue stems from the fact that training data not only contains intended correlations for the model to learn but also unintended biased correlations. A notable example is the COMPAS algorithm [4], widely employed for recidivism prediction, which inadvertently adopted biased correlations between African American individuals and recidivism, resulting in unjust sentencing decisions based on racial features.

Based on whether they align with biased correlations, data samples can be divided into *bias-aligned* (for instance, African American recidivism) and *bias-conflicting* samples. The essence of debiasing lies in balancing the influence of these two types of samples on the model. Conventional debiasing methods [10, 24, 33] either necessitate bias annotations for each training sample, distinguishing between bias-aligned and bias-conflicting samples, or utilize prior knowledge of bias types to construct specialized debiasing networks. However, acquiring known bias information is often impractical in real-world scenarios, as it entails extensive experimentation and labor-intensive bias label annotation. Hence, the research focus in debiasing is shifting towards the more practical approach of unsupervised debiasing [14, 18], capable of bias reduction without bias label annotation.

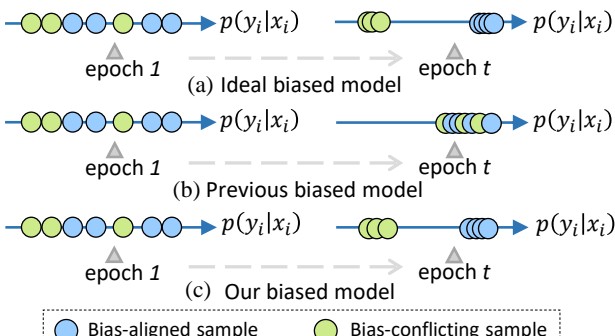

**Figure 1: (a) Ideal biased model. (b) The previous biased model fits the target label. (c) Our biased model differentiates between bias-aligned and bias-conflicting samples. $p(y_i|x_i)$ represents the probability of being classified as the target label.**

Unsupervised debiasing typically involves two main stages: (i) training a biased model using target labels to uncover data bias and (ii) utilizing the bias information revealed by the biased model for debiasing. The biased model is expected to fit well with bias-aligned examples and poorly with bias-conflicting examples to distinguish between these two types of samples, as shown in Fig. 1(a). However, contrary to expectations, our investigation into the efficacy of previous biased models revealed that these models struggled to rely on bias features, gradually fitting target labels instead, as illustrated in Fig. 1(b). Consequently, this inability to effectively uncover the bias directly results in a subpar debiasing performance by the target model.

The debiasing problem thus transfers to how to ensure the biased model relies on the bias features. Data poisoning, a method that aims to poison training samples so that the model learns attackers' malicious solutions, has recently drawn massive attention [2, 12]. Inspired by this, we propose a poisoning framework called *Iterative Poisoning of Bias-Conflicting Samples*, which iteratively embeds the bias rules learned by the biased model into the poisoned training data by altering the target labels. The objective is that the biased model, when trained using this poisoned data, only learns the bias rules, devoid of influence from target labels. Following this, we formally propose a novel debiasing method called *Poisoner*, which employs guided data poisoning to iterative poison bias-conflicting samples. Specifically, given that the early biased model has relied on some bias features, we can *identify* some bias-conflicting samples by observing the response of the sample to the bias rule in the representation space. Then, we *poison* the identified bias-conflicting samples with error-minimizing label poisoning to save the bias rule that the model has currently learned. Finally, we *update* the biased model in the poisoned training data to ensure that the model continues to learn new biases. These three steps are executed *iteratively* to continuously accumulate more bias in the model and refine the model's ability to identify bias-conflicting samples.

Benefiting from the accumulation of biases through data poisoning, the biased model comprehensively learns the biases present in the data, as illustrated in Fig. 1(c). If the biased model can correctly classify a sample, then this sample is bias-aligned; otherwise, it is bias-conflicting. Leveraging pseudo-labels for bias features,

we amplify the training weight of bias-conflicting samples to balance the target model's learning between bias-aligned and bias-conflicting samples. However, bias-conflicting samples assigned excessive weight may quickly become overfitted in the target model, diminishing their effectiveness in influencing the model's learning. To mitigate this issue, we introduce the *Group-wise Inverse Focal Loss* to enhance the model's focus on overfitted bias-conflicting samples. Consequently, the target model is prevented from learning the bias rules present in the data, which promotes an unbiased learning process.

We summarize our main contributions as follows:

- We propose *Poisoner*, a novel unsupervised debiasing method that employs poisoning to expose the potential bias in the data and eliminates the bias via group reweighing.
- We introduce a guided poisoning mechanism that encourages the biased model to specifically fit the bias features. This approach opens up new possibilities for the benign application of data poisoning.
- We perform extensive experiments on commonly used benchmarks, and our proposed method achieves state-of-the-art performance in both fairness and accuracy.

## 2 Motivation and Justification

### 2.1 Preliminary

Unsupervised debiasing methods involve the two models: the biased model and the target model. The biased model is trained on a dataset that solely contains the target label, and aims to subtly perceive the bias within the data. Subsequently, the target model performs debiasing based on the bias captured by the biased model. Evidently, the quality of the bias exposed by the biased model determines the performance of the target model's debiasing.

Two prevalent choices exist for a bias model: an Empirical Risk Minimization (ERM) model or a Generalized Cross Entropy (GCE) model. Liu et al. [21, 27] leverage the cross-entropy loss to train an ERM model as a biased model. Nam et al. [18, 23] consider that samples that are easy to classify are more likely to be samples aligned with bias, and therefore propose to employ a GCE loss [38] that pays more attention to samples that are easier to classify to train the biased model. The GCE loss is defined as:

$$\text{GCE}(p(x;\theta), y) = \frac{1 - p_y(x;\theta)^q}{q} \qquad (1)$$

where $p_y$ is the softmax probability output for the target (train) label $y$, and $q \in (0, 1]$ is a constant that controls the degree of amplification of attention to samples that are easy to classify.

### 2.2 Biased Models Fail to Fully Rely on Bias

For an ideal biased model, we expect it to fit well with bias-aligned examples and poorly with bias-conflicting examples to distinguish between these two types of samples. We train ERM model and GCE model over 50 epochs using commonly used datasets CelebA [22]. The target labels and the bias features are big-nose and gender, respectively. We document the error rate of the biased models on each group of training data after each epoch in Fig. 2(a) and Fig. 2(b). During the training process of ERM models, the error rates for all training data, including the bias-conflicting samples, are relatively

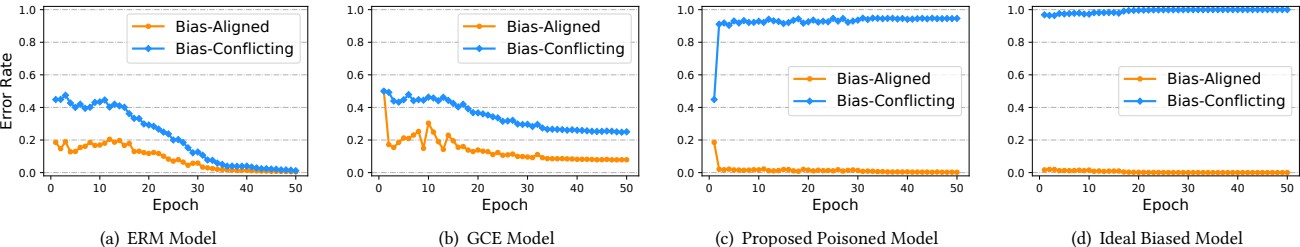

Figure 2: Classification error rates of different biased models on the training data of CelebA: (a) ERM Model, (b) GCE Model, (c) Proposed Poisoned Model (assumed with known bias-conflicting samples), and (d) Ideal Biased Model.

low, decreasing to nearly zero in the end. Although the error rate of bias-conflicting samples in the GCE model does not approach zero, the error rates at all moments throughout the entire training process are also relatively low, and show a decreasing trend. This indicates that the biased model fits the target label and fails to fully rely on bias.

## 2.3 Data Poisoning for Exposing Bias

**Early models are biased but not enough.** In the early stages of ERM model and GCE model training, as illustrated in Fig. 2(a) and Fig. 2(b), the error rate for bias-conflicting samples is significantly higher than that for bias-aligned samples. This suggests that early models tend to rely heavily on bias features for decision-making. However, this is far from sufficient. For instance, at epoch 2, ERM model can detect 46% of the bias-conflicting samples (with a 46% error rate for bias-conflicting samples). The already scarce bias-conflicting samples are further reduced by 54%.

**Error-minimizing label poisoning.** Data poisoning, which aims to poison training samples so that models learn attackers' malicious intent, has recently drawn massive attention [2, 12]. Inspired by this, we propose the following idea: *Can we leverage data poisoning to further reinforce biases that early models have already learned?* This would ensure that the model, when trained on poisoned data, will further learn these biases.

We propose *Error-minimizing Label Poisoning* to save the model's bias by poisoning the target label $y$ to the poisoned label $\hat{y}$:

$$\min_{\hat{y}} \mathcal{L}\left(f_B(\boldsymbol{x}; \theta), \hat{y}\right) \tag{2}$$

where $\mathcal{L}$ is cross-entropy loss. $\theta$ denotes the parameter of the model $f_B$. $\hat{y}$ is assigned as the predicted label of the model ensuring error-minimizing. By minimizing errors, the samples $(x, \hat{y})$ align with the bias rules that the model has already learned.

However, this may present two issues. Firstly, due to the model's limited reliance on bias features at early stages, the poisoned data may only introduce a subset of the existing bias rules. For instance, when poisoning at Epoch 2 of the ERM model on CelebA, only the labels of 46% of the bias-conflicting samples were altered to new labels that conformed to the bias rules. Secondly, there is a possibility of noise, where label changes may not be solely attributed to the bias rules learned by the model. For instance, the labels of approximately 20% of the bias-aligned samples were modified to incorrect labels.

**Iterative Poisoning Bias-conflicting samples.** To address the issue of the early model's limited reliance on bias features, we propose an iterative poisoning scheme to accumulate the biases learned by the model. To mitigate the noise issue in poisoning, we suggest focusing solely on poisoning the bias-conflicting samples. Bias-conflicting samples inherently contradict the bias, thus even if noise is introduced during the poisoning process, it will not undermine the bias rules within the data. This integration of ideas, termed *Iterative Poisoning Bias-conflicting samples*, can be formally expressed as:

$$\arg\min_{\theta} \mathbb{E}_{(\boldsymbol{x}, \hat{y}) \sim \hat{\mathcal{D}}} \left[ \min_{\hat{y}, \text{ if } x \in \mathcal{D}_c} \mathcal{L}\left(f_B(\boldsymbol{x}; \theta), \hat{y}\right) \right] \tag{3}$$

Here, $\hat{\mathcal{D}}_c \subset \hat{\mathcal{D}}$ represent the subsets of bias-conflicting samples. $\hat{y}$ denote the modifiable label. The inner *min* serves to save model bias to poisoned data by poisoning only the bias-conflicting samples $x \in \hat{\mathcal{D}}_c$, while the outer *min* operation serves to accumulate more model bias.

To validate the idea, we operate *Iterative Poisoning Bias-conflicting samples* under the assumption of prior knowledge regarding which samples are bias-conflicting. However, it's important to note that this assumption is solely for validation purposes; in reality, we do not know which samples are bias-conflicting, and we propose a method to identify bias-conflicting samples in Section 3.2. The outcomes of *Iterative Poisoning Bias-conflicting samples* are illustrated in Fig. 2(c). Notably, the bias-conflicting samples are almost entirely misclassified, as indicated by the error rate calculated based on the original clean labels. Furthermore, we directly trained the model using bias labels rather than target labels to simulate the scenario of an ideal biased model, as depicted in Fig.2(d). Comparing Fig.2(c) with Fig.2(d) shows that our proposed approach achieves performance similar to that of the ideal biased model in later stages.

## 3 Method

### 3.1 Overview

We propose a novel debiasing method, referred to as the *Poisoner*, depicted in Fig. 3. This method incorporates two models. The first is a biased model, $f_B$, which is designed to uncover potential data bias. The second is a target model, $f_T$, which performs debiasing operations based on the bias identified by $f_B$.

**Biased model.** Drawing inspiration from previous analyses, we propose *guided data poisoning* to ensure that the biased model is dependent on bias features. This will be provided in Section 3.2.

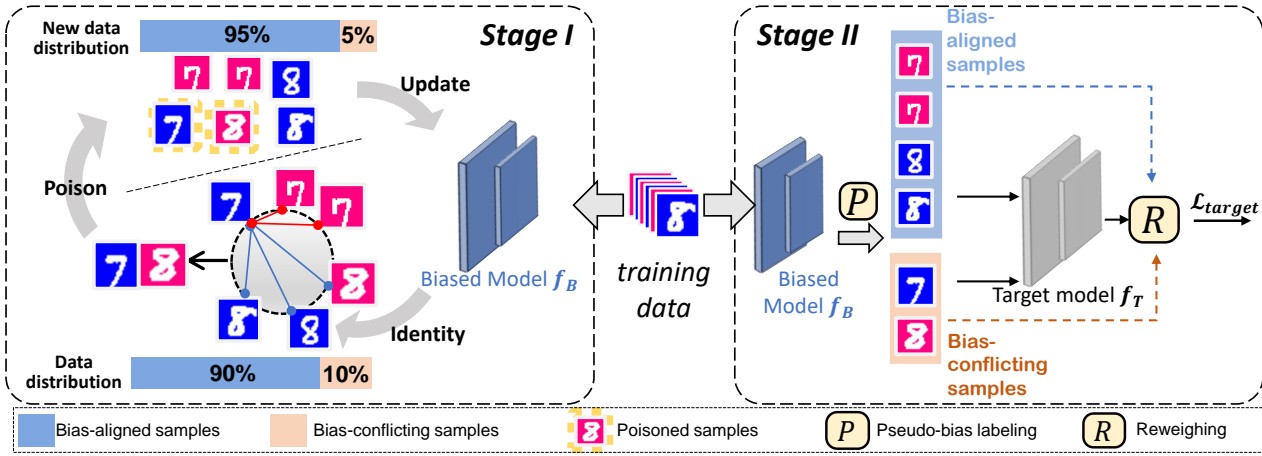

**Figure 3: Illustration of our proposed method *Poisoner*. Stage I: Training an auxiliary biased model $f_B$ with guided data poisoning to differentiate between bias-aligned and bias-conflicting samples (Sec. 3.2). Stage II: Learning a debiased target model $f_T$ with sample reweighing (Sec. 3.3).**

**Target model.** We *reweight* the training weight of bias-conflicting samples to balance the influence of bias-aligned and bias-conflicting samples for debiasing. We also propose Group-wise Inverse Focal Loss to mitigate potential challenges arising from overfitting bias-conflicting samples. This will be discussed in Section 3.3.

## 3.2 Exposing Bias via Guided Data Poisoning

Section 2 demonstrated that the previous biased models failed to fully rely on bias. The idea of iteratively poisoning only the bias-conflicting samples has been validated as a solution to this problem. In this section, we introduce Guided Data Poisoning, which is designed to guide the iterative poisoning of bias-conflicting samples.

**Identify.** A model trained on a biased dataset tends to rely on bias features of samples for prediction. This leads to samples with identical target features but different bias features being separated in the representation space, and samples with similar bias features but different target features being brought closer together [10, 24].

This insight guides us to use similarity as a metric for identifying bias-conflicting samples. More specifically, considering that the majority of samples within each target class are aligned with these biases, and the samples that conflict with these biases deviate from these aligned samples and may even approach other target classes, we can calculate whether a sample is far from samples of the same class and close to samples of other classes to identify whether it is a bias-conflicting sample. We adopt the main idea of supervised contrastive loss [15]. For each sample $x_i$ in the current batch, we compare $x_i$ with the other data in the mini-batch $\mathcal{B}$ as follows:

$$d_{\text{con}}(x_i; \theta) = -\frac{1}{|J(i)|} \sum_{j \in J(i)} log \frac{\exp(z_i \cdot z_j)}{\sum_{a \in A(i)} \exp(z_i \cdot z_a)} \quad (4)$$

where $z_i = f_B^{-2}(x_i)/||f_B^{-2}(x_i)||$ is a normalized feature of sample $x_i$ extracted from the penultimate layer $f_B^{-2}$ of biased model $f_B$. $J(i)$ denotes the index set of samples in the current batch that have the same target label $\text{label}^{t-1}(x_i)$ as $x_i$. $A(i)$ represents the index set of all samples in the current batch $\mathcal{B}$, excluding $x_i$. Furthermore,

batch $\mathcal{B}$ is obtained through class-balanced sampling because we aim to consider all target classes fairly.

The larger the $d_{\text{con}}(x_i; \theta)$, the more likely it is that the sample $x_i$ is a bias-conflicting sample of the current target class. Then, we compute the $(100 - p)_{th}$ percentile within the $\{d_{\text{con}}(x_i; \theta)\}_{i=1}^{|\mathcal{B}|}$, denoted as $q_p$, where $|\mathcal{B}|$ is the size of mini-batch $\mathcal{B}$. $p \in (0, 100)$ is a hyper-parameter that controls the proportion of bias-conflicting samples selected in a single batch. Subsequently, the indicator of whether a sample is a bias-conflicting sample can be formalized as:

$$\text{conflict}(x_i) = \begin{cases} 1 & \text{if } d_{\text{con}}(x_i; \theta) > q_p \\ 0 & otherwise \end{cases} \quad (5)$$

In other words, $x_i$ is considered a bias-conflicting sample if $d_{\text{con}}(x_i; \theta)$ is greater than $q_p$.

**Poison.** At the beginning of the training of the biased model, each sample's target label is initially the clean label without poisoning, *i.e.*, $\text{label}^{t=0}(x) = y$, and $y$ is the clean label. During the training step $t$, we selectively poison the label of the samples that are identified as bias-conflicting (*i.e.*, conflict $(x) =1$) for error-minimizing $\min_{\hat{y}} \mathcal{L}(f_B(x), \hat{y})$:

$$\text{label}^t(x_i) = \begin{cases} \text{label}^{t-1}(x_i) & \text{if } \text{conflict}(x_i) = 0 \\ Y(f_B(x_i)) & \text{if } \text{conflict}(x_i) = 1 \end{cases} \quad (6)$$

where $\text{label}^t$ is the poisoned label in iteration $t$. $Y(f_B(x_i))$ is the predicted label of biased model in current iteration $t$. By changing the label of the bias-conflicting sample $x_i$ to $Y(f_B(x_i))$, the biases within the data are further reinforced.

**Update.** During training step $t$, we use all samples in the current mini-batch $\mathcal{B}$ and their corresponding poisoned labels $\text{label}^t$ as the training data for the biased model $f_B$. The cross-entropy loss is employed for training as follows:

$$\arg\min_{\theta} \mathbb{E}_{(x) \sim \mathcal{D}} \mathcal{L}_{CE}(f_B(x; \theta), \text{label}^t(x)) \quad (7)$$

**Iterate.** We iteratively repeat the steps of *Identify*, *Poison*, and *Up-*

---

**Algorithm 1** Training of the biased model

---

1: **Input**: dataset $\mathcal{D} = \{(x_i, y_i)\}_{i=1}^{n}$, where the label $y_i$ can be modified; parameter $\theta$ of model $f_B$; number of training steps $T$; batch size $m$; hyperparameter $p$.
2: **Output**: parameter $\theta$ of model $f_B$
3: **for** t = 1,…,$T$
4:   Sample a class-balanced batch $\mathcal{B} = \{(x_j, y_j)\}_{j=1}^{m}$ from $\mathcal{D}$
5:   /** *Identify bias-conflicting examples* **/
6:   Calculate $d_{\text{con}}$ for all samples in $\mathcal{B}$, using Eq. 4
7:   Get the bias-conflicting indicator (conflict($x$)), using Eq. 5
8:   /** *Poison bias-conflicting examples* **/
9:   **for** j = 1,…,$m$
10:     **if** $conflict(x_j) = 1$
11:       Get the predicted label $Y(f_B(x_j))$ of $f_B$
12:       Modify the label of $x_j$ in $\mathcal{D}$ to $Y(f_B(x_j))$
13:       Modify $y_j$ in $\mathcal{B}$ to $Y(f_B(x_j))$
14:   /** *Update biased model* $f_B$ **/
15:   $\theta \leftarrow \theta - \nabla\mathcal{L}_{CE}(\mathcal{B})$

---

*date*. In this iterative process, the biased model gradually becomes more reliant on bias features. The pseudo-code of the training of the biased model is presented in Alg. 1.

## 3.3 Debiasing via Group Reweighing

Upon completing the training phase for the biased model $f_B$, the model's output can be used to distinguish between bias-aligned and bias-conflicting samples. A sample $x_i$ is considered bias-aligned if it is classified correctly, meaning its output aligns with the clean label $y_i$. Conversely, a sample is labeled as bias-conflicting if its classification contradicts the clean label.

**Group Reweighing.** The primary source of model bias is the biased correlations inherent in the training data. To counteract this correlation and train an unbiased target model, we employ sample reweighing to rebalance the influence of samples that exhibit varying bias features. Specifically, the training weights of bias-conflicting and bias-aligned samples are adjusted as follows:

$$w\left(x_i\right) = \begin{cases} \frac{|\mathcal{D}|}{|\mathcal{D}_a|} & \text{if } x_i \in \mathcal{D}_a \\ \lambda \cdot \frac{|\mathcal{D}|}{|\mathcal{D}_c|} & \text{if } x_i \in \mathcal{D}_c \end{cases} \tag{8}$$

Here, $\mathcal{D}_a \subset \mathcal{D}$ and $\mathcal{D}_c \subset \mathcal{D}$ represent the subsets of bias-aligned and bias-conflicting samples, respectively, partitioned from the overall dataset $\mathcal{D}$. The symbols $|\mathcal{D}|$, $|\mathcal{D}_a|$ and $|\mathcal{D}_c|$ denote the number of samples in these sets. The hyperparameter $\lambda$ controls the degree of additional emphasis given to bias-conflicting samples, typically ranging between 1 and 1.5. Since some bias-conflicting samples may not be correctly identified by $f_B$, even though they are few in number, we need to emphasize bias-conflicting samples additionally. Our subsequent experiments demonstrate that the parameter $\lambda$ is not sensitive.

Then, the loss function of the target model $f_T$, which is trained with weighted samples, can be formalized as follows:

$$\mathcal{L}_{target} = \mathbb{E}_{(x,y)\sim\mathcal{D}} \, w(x) \cdot \mathcal{L}_{CE}\left(f_T(x), y\right) \tag{9}$$

**Group-wise Inverse Focal Loss.** However, assigning a larger training weight to the already scarce bias-conflicting samples can quickly lead to the target model overfitting these samples, as shown in Fig. 6(a). That is, the probability of the target label $y$, denoted as $p_y$, tends to 1, resulting in the loss approaching 0. As a result, the contributions of bias-conflicting samples to the training of the target model become limited, hindering the debiasing.

To address the negative effects of overfitting, we introduce the concept of group-wise inverse focal loss, which aims to more focus learning on overfitted bias-conflicting samples. To tackle the long-tail problem, *Focal Loss* [20] applies a dynamic scaling factor to the cross entropy loss to focus learning on hard misclassified examples, represented as $\mathcal{L}_{FL} = -\left(1 - p_y\right)^{\gamma} \log\left(p_y\right), \gamma > 0$, where the scaling factor $(1 - p_y)^{\gamma}$ decays to zero as confidence $(p_y)$ in the correct class increases. In contrast, we aim to enhance focus learning on easy misclassified (*i.e.*, overfitted) examples, thus we invert the Focal Loss to $\mathcal{L} = -\left(1 - p_y\right)^{-\gamma} \log\left(p_y\right)$. However, this will not directly prompt the model to focus on bias-conflicting samples due to some bias-aligned samples also having a high $p_y$. To this end, we propose Group-wise Inverse Focal Loss:

$$\mathcal{L}_{GIFL}\left(x, y\right) = \begin{cases} -\left(1 - \overline{p_y^a}\right)^{-\gamma} \log\left(p_y\right) & \text{if } x \in \mathcal{D}_a \\ -\left(1 - \overline{p_y^c}\right)^{-\gamma} \log\left(p_y\right) & \text{if } x \in \mathcal{D}_c \end{cases} \tag{10}$$

Here, $p_y$ represents the probability that the target model $f_T$ predicts sample $x$ as the correct class $y$. $\overline{p_y^a}$ and $\overline{p_y^c}$ respectively denote the average $p_y$ values on bias-aligned and bias-conflicting samples in the current mini-batch. The focusing parameter $\gamma > 0$ smoothly adjusts the rate at which overfitted examples are prioritized (we found that $\gamma$ approaching 1 works best in our experiments). If bias-conflicting samples are overfitted (resulting in high $\overline{p_y^c}$ values), they will be assigned a larger loss, directing the target model to pay more attention to these samples during training.

Moreover, for numerical stability of $\mathcal{L}_{GIFL}$, we introduce an adaptive multiplicative factor $\beta = 1/\left(\left(1 - \overline{p_y^c}\right)^{-\gamma}\right)$. Thus, the overall training loss for the target model is given by:

$$\mathcal{L}_{target} = \mathbb{E}_{(x,y)\sim\mathcal{D}} \, w(x) \cdot \beta \cdot \mathcal{L}_{GIFL}\left(x, y\right) \tag{11}$$

## 4 Experiment

## 4.1 Experimental Settings

**Datasets.** We construct experiments on eight debiasing tasks across five benchmark datasets. **CelebA**, an industrial-scale dataset, contains about 200k facial images with 40 binary attribute annotations. We select target attributes that display the high Pearson correlation with gender, and perform gender debiasing for the recognition of each target attribute. The selected target attributes are *bignose*, *attractive*, *blonde*, and *bags-under-eyes*. We also use the **Waterbirds** [25] dataset, where waterbirds and land birds are highly correlated with wet and dry backgrounds, respectively. The proportion of bias-aligned samples that conform to this biased correlation is 95%. Our objective is to eliminate the influence of the background on bird recognition. The **Dogs&Cats** [16] dataset contains a fur color bias. Each animal species has a correlation of 0.95 with a specific fur color. Our goal is to eliminate the influence of the animal's

**Table 1: The Model Bias (in %, Equalodds, ↓), Avg. Group Accuracy (in %, ↑), and Worst Group Accuracy (in %, ↑) of models trained on CelebA. Here *bn, a, bl* and *bu* respectively denote *bignose, attractive, blonde, bags-under-eyes*. The best results with unknown biases are highlighted in bold. \* indicates that the method knows the bias label of training samples.**

| Method | T=*bn* | | | T=*a* | | | T=*bl* | | | T=*bu* | | | Avg. | | |
|---|---|---|---|---|---|---|---|---|---|---|---|---|---|---|---|
| | Model Bias ↓ | Avg ACC ↑ | Worst ACC ↑ | Model Bias ↓ | Avg ACC ↑ | Worst ACC ↑ | Model Bias ↓ | Avg ACC ↑ | Worst ACC ↑ | Model Bias ↓ | Avg ACC ↑ | Worst ACC ↑ | Model Bias ↓ | Avg ACC ↑ | Worst ACC ↑ |
| Vanilla | 31.40 | 71.01 | 43.36 | 23.51 | 74.13 | 62.34 | 32.05 | 82.83 | 58.32 | 17.70 | 71.51 | 43.71 | 26.17 | 74.87 | 51.93 |
| Focal | 23.29 | 70.99 | 47.61 | 24.25 | 76.56 | 61.36 | 30.05 | 83.78 | 64.35 | 16.31 | 70.87 | 42.47 | 23.48 | 75.55 | 53.94 |
| LfF | 16.73 | 68.42 | 53.12 | 17.53 | 76.57 | 67.36 | 29.75 | 75.32 | 49.10 | 18.73 | 70.53 | 43.51 | 20.69 | 72.71 | 53.27 |
| JTT | 14.29 | 72.31 | 55.09 | 15.06 | 77.61 | 65.33 | 13.07 | 85.02 | 75.53 | 15.34 | 70.22 | 54.01 | 14.44 | 76.29 | 62.49 |
| DebiAN | 29.03 | 69.41 | 39.75 | 22.39 | 76.63 | 59.22 | 29.35 | 77.29 | 63.81 | 19.38 | 70.65 | 44.95 | 19.38 | 70.65 | 44.95 |
| Echoes | 19.95 | 66.19 | 42.73 | 27.52 | 72.57 | 65.11 | 18.27 | 76.53 | 63.52 | 16.52 | 70.52 | 51.23 | 20.57 | 71.45 | 55.64 |
| BE | 15.57 | 69.57 | 56.54 | 16.21 | 76.74 | 67.14 | 15.84 | 80.58 | 69.21 | 14.32 | 71.65 | 59.14 | 15.48 | 74.63 | 63.00 |
| Poisoner | **6.56** | **74.49** | **69.61** | **3.57** | **79.98** | **77.04** | **8.05** | **91.37** | **85.78** | **7.61** | **76.58** | **68.29** | **6.44** | **80.61** | **75.18** |
| GroupDRO\* | 5.54 | 74.97 | 66.28 | 4.02 | 79.72 | 76.83 | 7.61 | 92.51 | 81.97 | 7.96 | 77.81 | 67.00 | 6.28 | 81.25 | 73.02 |

fur color on animal recognition. For **C-MNIST** [17], the task is to recognize digits (0-9), in which the images of each target class are dyed by the corresponding color with probability $\rho$, while the remaining samples are randomly colored with other colors. For the version of C-MNIST[1] we use, the probability $\rho$ is 0.99. For C-MNIST[2], the probability $\rho$ is 0.98. Our goal is to eliminate color bias. Lastly, the **ImageNet-B** [36] dataset offers a complex and realistic testbed, featuring ten types of natural noise patterns as bias attributes. Each target class exhibits a correlation of 0.95 with a specific type of natural noise. Our ambition within this dataset is to counteract the influence of natural noise on object recognition.

**Metric.** We aim to answer two main questions: (1) How does the fairness of *Poisoner* compare to other methods? (2) How does the classification performance of *Poisoner* compare with other methods? To answer the first question, we examine whether the accuracy of the model predictions changes with shifts in bias attributes (*e.g.*, gender). We use **Equalodds** [8] to measure fairness. For instance, the measure of gender fairness on the CelebA dataset is as follows:

$$\frac{1}{|Y|} \sum_y \left| \Pr_{b^0}(\tilde{Y} = y \mid Y = y) - \Pr_{b^1}(\tilde{Y} = y \mid Y = y) \right| \quad (12)$$

where $Y$ denotes target labels, $\tilde{Y}$ denotes outputs of models, and $b^0$ and $b^1$ represent different groups in terms of bias attributes such as *male* and *female*. Considering that Colored-MNIST and ImageNet-B not only have two types of bias features, we use the difference between the average accuracy and the worst group accuracy on each target class as model bias.

To answer the second question, we first divide the test set into different groups based on the bias attributes and target attributes. Then, we report two types of accuracy: the **Average Group Accuracy** and the **Worst Group Accuracy**.

**Baselines.** To evaluate the effectiveness of our method, we compare it against prior methods including *Vanilla, Focal loss* [20], *LfF* [23], *JTT* [21], *DebiAN* [19], *Echoes* [11], *BE* [18], *GroupDRO* [25]. The *Vanilla* model, trained solely with the original cross-entropy loss, employs no debiasing strategies. *BE* is a method that relies on *LfF* or other debiasing methods, and we use it in conjunction with *LfF*. *GroupDRO* explicitly leverages the bias labels (e.g., the gender

labels) during the training phase, while others require no prior knowledge of the biases.

**Implementation.** Following previous research, we utilize a multi-layer perceptron (MLP) with three hidden layers for the Colored MNIST dataset. For ImageNet-B, we use ResNet-34 [9]. For other datasets, we employ ResNet-18 [9]. As for the parameter $P$ in our method, which controls the percentage of samples identified as bias-conflicting in each training step of the biased model, we set it to 10 for CelebA and 5 for the other datasets. We set the hyperparameter $\gamma$ to 1 for all experiments. We use the Adam optimizer for all baselines with a learning rate of 1e-3.

### 4.2 Main Results

**Debiasing with social bias.** Table 1 shows the model bias, average group accuracy, and the worst group accuracy of models on four target tasks of CelebA, with gender as the bias attribute. *Vanilla* models record severe model bias as they are optimized to capture the biased statistical properties of training data without any constraints. Various debiasing methods demonstrate different levels of effectiveness in bias mitigation. In comparison with other unsupervised debiasing methods (which do not require prior knowledge regarding bias), the proposed *Poisoner* achieves the best performance in terms of fairness and accuracy across all four target tasks. For instance, for the big-nose recognition task (T=*bn*), Vanilla models display significant fairness issues (model bias = 31.40%), with the worst group accuracy being only 43.36%. Our proposed poisoner reduces the model bias to 6.56% and improves the worst group accuracy to 74.49%. The debiasing performance of other methods is limited in comparison. Our method's state-of-the-art performance in unsupervised debiasing confirms our biased models precisely identify bias-conflicting samples. Furthermore, we also compared our method with *GroupDRO*, which requires the use of bias labels in training. Compared to *GroupDRO*, our method achieves competitive results, even achieving better performance in T=*a* task.

**Debiasing with general bias.** Beyond the social bias that is central to fairness research, we have also benchmarked our method against other methods on four datasets that encompass general

**Table 2: The debiasing performance on four benchmark datasets with general bias. * indicates that the method knows the bias label of training samples. - denotes that the test set is not applicable for evaluating model bias.**

| Method | WaterBirds | | | Dogs & Cats | | | C-MNIST[1] | | | C-MNIST[2] | | | ImageNet-B | | |
|---|---|---|---|---|---|---|---|---|---|---|---|---|---|---|---|
| | Model Bias ↓ | Avg ACC ↑ | Worst ACC ↑ | Model Bias ↓ | Avg ACC ↑ | Worst ACC ↑ | Model Bias ↓ | Avg ACC ↑ | Worst ACC ↑ | Model Bias ↓ | Avg ACC ↑ | Worst ACC ↑ | Model Bias ↓ | Avg ACC ↑ | Worst ACC ↑ |
| Vanilla | 35.71 | 78.17 | 46.61 | - | 50.35 | 47.76 | 43.27 | 56.28 | 10.30 | 32.73 | 67.09 | 7.02 | 67.09 | 61.37 | 23.60 |
| Focal | 30.12 | 77.99 | 56.71 | - | 68.25 | 65.85 | 43.82 | 56.91 | 3.05 | 29.61 | 70.10 | 9.01 | 66.99 | 61.16 | 19.30 |
| LfF | 13.57 | 79.64 | 63.56 | - | 72.91 | 50.10 | 12.99 | 75.26 | 28.41 | 11.29 | 83.92 | 59.57 | 37.97 | 64.21 | 28.20 |
| JTT | 12.06 | 75.61 | 58.19 | - | 73.95 | 67.45 | 13.07 | 74.05 | 30.03 | 13.38 | 78.34 | 56.10 | 35.24 | 65.38 | 30.54 |
| DebiAN | 12.36 | 77.72 | 59.22 | - | 71.24 | 67.68 | 17.08 | 70.48 | 36.18 | 15.25 | 79.51 | 57.61 | 44.58 | 62.40 | 29.32 |
| Echoes | 14.52 | 78.79 | 62.73 | - | 84.56 | 82.17 | 16.27 | 79.18 | 36.28 | 13.48 | 78.42 | 57.23 | 38.64 | 62.54 | 27.91 |
| BE | 13.21 | 76.74 | 67.14 | - | **85.59** | 79.21 | 14.98 | 81.39 | **39.41** | 11.18 | 85.66 | 59.14 | 42.61 | 64.72 | 28.96 |
| Poisoner | **3.57** | **84.26** | **78.01** | - | 84.81 | **83.02** | 12.98 | 82.57 | 37.40 | **8.77** | **87.51** | **64.75** | **31.51** | **66.28** | **32.43** |
| GroupDRO* | 5.23 | 86.72 | 79.83 | - | 81.53 | 68.47 | 16.51 | 83.13 | 29.61 | 10.58 | 85.03 | 42.70 | 29.22 | 67.76 | 44.32 |

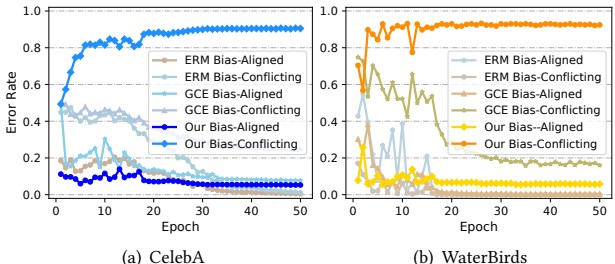

Figure 4: The classification error rate of GCE model and our biased model on the training data of CelebA and WaterBirds.

biases. The results, as depicted in Table 2, demonstrate that our method consistently achieves superior debiasing performance. This underscores that the effectiveness of our method is not contingent on the type of bias attribute.

### 4.3 The Efficacy of Our Biased Model in Uncovering Bias

An ideal biased model should have a lower error rate for bias-aligned samples and a higher error rate for bias-conflicting samples in the training data, thus facilitating the delineation of pseudo-labels for bias features. Initially, we demonstrate the comparative advantages of our biased model over the ERM and GCE models, as illustrated in Fig. 4. As training progresses, our method gradually separates bias-aligned and bias-conflicting samples. Lastly, the bias-conflicting samples are almost entirely misclassified, indicating that these samples have been almost wholly mined. In contrast, both ERM and GCE fit bias-conflicting samples and fail to mine them.

Furthermore, we introduce a metric called `Bias Accuracy` to quantitatively assess the biased model's efficacy in exposing biases. Specifically, Bias Accuracy is calculated as the average of the accuracy and error rate of the biased model on bias-aligned and bias-conflicting samples, respectively. We present the `Bias Accuracy` achieved by various biased models on different datasets in Table 3. It is evident that our biased model significantly outperforms alternative methods in exposing biases. Moreover, our biased model exhibits overwhelming superiority for tasks characterized by lower

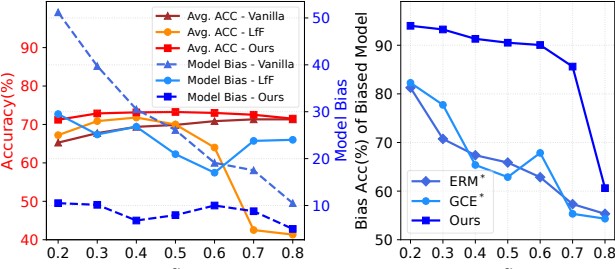

Figure 5: The debiasing performance and the capability to uncover bias under different degrees of data balance ($\alpha$). Larger $\alpha$ indicates that the training set is more balanced.

degrees of dataset bias (*i.e.*, biases are less apparent), such as the four tasks within the CelebA dataset. This observation suggests the applicability of our approach not only in scenarios characterized by high levels of bias intensity but also in contexts where biases are deeply embedded and less discernible.

### 4.4 Controlled Experiments on Bias Intensity

Real-world datasets often exhibit varying degrees of bias. To demonstrate the versatility of our method in scenarios with less extreme bias, we manipulate the bias intensity in the CelebA training data by removing some samples. Fig. 5 (left) illustrates our method's debiasing performance under different data bias scenarios, with the degree of data balance $\alpha$ representing the ratio of bias-conflicting samples to bias-aligned samples. A higher $\alpha$ indicates lower data bias intensity. Our method shows significant bias mitigation across all degrees of data balance, indicating its effectiveness in both extreme and less pronounced bias scenarios. Conversely, the GCE-Based *LfF* method fails to mitigate bias for high data balance scenarios ($\alpha > 0.5$) and significantly decreases accuracy.

Moreover, we attribute this universal debiasing capability to the efficient uncovering of bias by our biased model across various degrees of data balance. Fig. 5 (right) displays the metric measuring the effectiveness of uncovering bias, the `Bias Accuracy`, with our biased model showing notable superiority over others. This further underscores the versatility of our approach.

**Table 3: The Bias Accuracy (in %, ↑) of different biased models. A higher value indicates stronger efficacy in exposing biases. * denotes early stopping applied at epoch 2.**

| Methods | CelebA_bn | CelebA_a | CelebA_bl | CelebA_bu | WaterBrids | Dogs&Cats | C-MNIST¹ | ImageNet-B |
|---------|-----------|----------|-----------|-----------|------------|-----------|----------|------------|
| ERM  | 51.10 | 50.75 | 51.36 | 55.21 | 50.00 | 55.43 | 65.70 | 59.84 |
| ERM* | 65.31 | 65.53 | 67.85 | 61.57 | 72.35 | 79.53 | 85.21 | 61.75 |
| GCE  | 58.53 | 57.35 | 59.57 | 58.96 | 60.31 | 72.17 | 59.50 | 84.73 |
| GCE* | 67.33 | 66.50 | 71.53 | 65.68 | 76.35 | 80.17 | 80.53 | 82.91 |
| Ours | **92.26** | **91.31** | **92.57** | **86.89** | **93.65** | **89.30** | **88.95** | **87.52** |

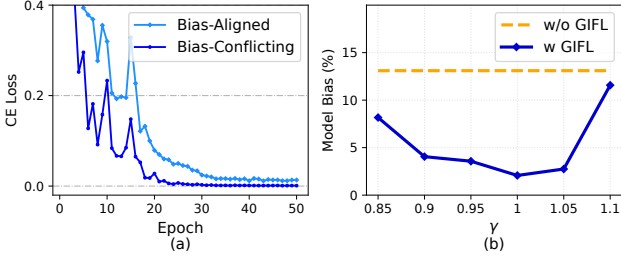

**Figure 6: Overfitting issues of bias-Conflicting samples and sensitivity to $\gamma$ on WaterBirds.**

**Table 4: The comparison of the results of our method with and without Group-wise Inverse Focal Loss (GIFL).**

| Metrics | GIFL | CelebA T=bn | Water Birds | Dogs & Cats | C-MNIST¹ | Imagenet -B |
|---------|------|-------------|-------------|-------------|----------|-------------|
| Avg. ACC | ✗ | 73.97 | 82.12 | 83.37 | 81.14 | 63.51 |
|          | ✓ | **74.49** | **84.26** | **84.81** | **82.57** | **66.28** |
| Model Bias | ✗ | 7.49 | 13.10 | - | 12.31 | 40.53 |
|            | ✓ | **6.56** | **3.57** | - | **12.98** | **31.51** |

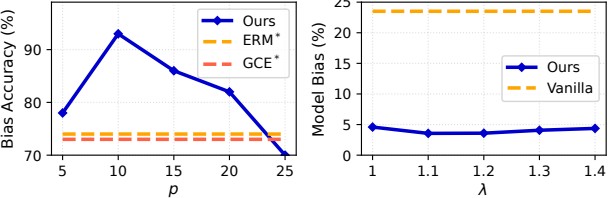

**Figure 7: Ablation on hyper-parameter $p$ and $\lambda$ on CelebA.**

### 4.5 Analysis

**Group-wise Inverse Focal Loss (GIFL) and $\gamma$.** We first demonstrate the overfitting issues of the target model on bias-conflicting samples under the condition of Group Reweighting without GIFL. In Fig. 6(a), the average cross-entropy (CE) losses on bias-conflicting and bias-aligned samples are presented. It can be observed that the loss on bias-conflicting samples remains consistently lower than that on bias-aligned samples, and quickly decreases to zero. This indicates that bias-conflicting samples are being overfitted, resulting in bias-aligned samples being the main influence during the target model training process. In Fig. 6(b), we illustrate the model bias of the target model with GIFL under different $\gamma$ values. It can be seen that as $\gamma$ approaches 1, the model bias decreases. We observe the

same trend on other datasets as well. Furthermore, Table 4 compares the results between the Poisoner with and without GIFL. Across all tasks, both the fairness and accuracy of the model degrade when GIFL is not applied. This underscores the effectiveness of GIFL in enhancing both the fairness and accuracy of the model.

**Different values of $p$ and $\lambda$.** We use $p \in (0, 100)$ to control the percentage of samples identified as bias-conflicting in each training round of the biased model. Fig. 7 illustrates the accuracy of bias identification for different values of $p$. When $p$ is relatively large, numerous bias-aligned samples are misclassified as bias-conflicting. This misidentification contradicts the original intent of guided data poisoning, resulting in noisy injections. Therefore, setting $p$ within the range of 5 to 20 is considered appropriate. We also examine debiasing performance with different $\lambda$ values. The results indicate that $\lambda$ exhibits insensitivity to debiasing performance.

## 5 Related work

**Bias Mitigation.** Early works on debiasing relied on prior knowledge about the biases present in the data (known as supervised debiasing). Some approaches [10, 24, 25, 32, 34, 35, 37] required explicit bias labels for each training sample, such as the gender of facial images. For instance, Zhang et al. [34, 35] employed adversarial training to minimize a discriminator's ability to predict bias labels, thus encouraging fair outputs.

Recent studies focus on the more realistic unsupervised debiasing: debiasing without any prior bias information. Unlike supervised debiasing, unsupervised debiasing first requires obtaining information about the biases. Typical approaches [17, 21, 23] assume that biased features are more readily learned by models compared to robust features. Thus, they train an auxiliary biased model that is expected to primarily rely on the biased features, considering the auxiliary model's outputs as pseudo-labels for bias. Some methods [11, 18] attempt to construct intentionally biased datasets to train the auxiliary biased model, while others utilize networks with limited capacity to uncover biases [26]. Furthermore, feature clustering [29] and fairness minimization [19] techniques have been employed to discover unknown biases.

**Poisoning Attack.** Poisoning attacks [2, 3, 7] in machine learning have emerged as a significant security concern, where adversaries manipulate training data to compromise the integrity and performance of models. Poisoning attacks involve injecting malicious data samples [13, 30], known as poisoned examples, into the training dataset. These examples are carefully crafted to deceive the learning algorithm and embed the attackers' malicious intent into the model's decision-making process. To counter poisoning attacks, various defense mechanisms have been developed [5, 6, 31].

## 6 Conclusion

In this study, we introduce *poisoner*, a novel unsupervised debiasing method. The *poisoner* couples guided data poisoning and model training to prompt the biased model to learn more bias continuously. Subsequently, it eradicates the bias uncovered by the bias model in the target model by assigning more attention to bias-conflicting samples. Experiments demonstrate its effectiveness in improving fairness and accuracy.

# Acknowledgments

This work is supported by the National Key R&D Program of China (No. 2023YFC3310700).

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
