# OpenReview forum: "Poisoning for Debiasing: Fair Recognition via Eliminating Bias Uncovered in Data Poisoning"
_acmmm.org/ACMMM/2024/Conference — MM2024 Poster_

### Official Review · Reviewer_Ldr7 · 2024-05-01

**Rating:** 4
**Confidence:** 2

**Summary:**

This paper proposed Poisoner, a novel unsupervised debiasing method that employs poisoning to expose the potential bias in the data and eliminates the bias via group reweighing. Extensive experiments verify its effectiveness.

**Strengths:**

1.The paper structure is OK. This paper introduces its motivation for proposing their approach, which can make the proposed scheme more convincing. Actually, I am not familiar with this topic, but the scheme seems OK to me.

2.The experimental results are comprehensive. The experimental results seem extensive and convincing for me.

3. The writing is good and easy to understand.

**Limitations:**

1. Confused descriptions. In line 164, the authors claim that they propose a unsupervised scheme. However, In lines 251-252, the authors propose Error-minimizing Label Poisoning approach, which seems contradictory to the unsupervised scenario.

2. Typos: Section 4.3 "The Efficacy of Our Biased Model in uncovering Bias" -> "The Efficacy of Our Biased Model in Uncovering Bias"

**Suitability:**

2

---

### Official Review · Reviewer_jGGe · 2024-05-24

**Rating:** 5
**Confidence:** 2

**Summary:**

This paper proposes a novel debiasing method based data poisoning method to embed the biases into the poisoned training data, thereby facilitating the training of a more effectively biased model, which is subsequently used to identify unbiased samples. The idea in this paper is straightforward with detailed explanations and sufficient experimental support.

**Strengths:**

1. This paper provides a well-articulated introduction and justification, making it easy for readers to comprehend the field of biased learning.

2.  The motivation and methodology presented are coherent, which utilize the correlation between the poison data-based attack and data bias to improve the biased model.

3. The experimental results presented in this paper are both comprehensive and extensive.

**Limitations:**

While I am not an expert in this field, I believe it would be more convincing to provide theoretical insights that analyze the benefits brought by the poisoned data.

**Suitability:**

2

---

### Official Review · Reviewer_yzp5 · 2024-05-24

**Rating:** 4
**Confidence:** 3

**Summary:**

Neural networks can suffer from bias issues, relying excessively on features that have spurious but strong correlations with target labels. This can lead to poor performance on data that does not follow these correlations. Traditional debiasing methods require labeling bias features, which is challenging and labor-intensive. Recent work tackles this by training two models: a bias model to identify bias features and a target model to eliminate the bias. However, these bias models often fit the target labels, failing to properly expose data bias.

To address this, the paper proposes a novel debiasing method called "poisoner" that utilizes data poisoning. It embeds the biases learned by the bias model into the training data, encouraging the models to learn more biases. This is achieved by coupling guided data poisoning and model training, which continuously prompts the bias model to identify and learn more biases. Then, samples that contradict these biased correlations are identified and their influence is amplified during the training of the target model, preventing it from learning such biased correlations. Experiments show that this method achieves superior debiasing performance.

**Strengths:**

1. This paper is well-written and easy to follow.

2. The experiments are sufficient and convincing to show the effectiveness of the proposed method.

3. I am not very familiar with this area, but this method is not trivial and seems novel.

**Limitations:**

1. More comparisons with existing methods such as Chroma-VAE are suggested.

2. It is better to clarify "label poisoning" in the title.


[1] @inproceedings{NEURIPS2022_80098914,
 author = {Yang, Wanqian and Kirichenko, Polina and Goldblum, Micah and Wilson, Andrew G},
 booktitle = {Advances in Neural Information Processing Systems},
 editor = {S. Koyejo and S. Mohamed and A. Agarwal and D. Belgrave and K. Cho and A. Oh},
 pages = {20351--20365},
 publisher = {Curran Associates, Inc.},
 title = {Chroma-VAE: Mitigating Shortcut Learning with Generative Classifiers},
 url = {https://proceedings.neurips.cc/paper_files/paper/2022/file/80098914b3b3bad79b80377751a85430-Paper-Conference.pdf},
 volume = {35},
 year = {2022}
}

**Suitability:**

2

---

### Official Review · Reviewer_RP5f · 2024-05-24

**Rating:** 3
**Confidence:** 2

**Summary:**

The paper titled "Poisoning for Debiasing: Fair Recognition via Eliminating Bias Uncovered in Data Poisoning" addresses the issue of biases in neural networks, which often rely on spurious correlations between features and target labels. Traditional debiasing methods require bias feature annotations, which are labor-intensive. Recent approaches use two models: a bias model to detect biases and a target model to mitigate them. However, these methods often fail as the bias model ends up fitting target labels instead of exposing biases.
To resolve this, the paper proposes a novel method called Poisoner, which employs data poisoning. This method embeds biases learned by biased models into poisoned training data, prompting the models to learn more biases. The process involves:
Guided Data Poisoning: Iteratively embedding bias rules into training data by altering target labels.
Model Training: Using poisoned data to ensure the biased model learns only biases.
Bias Identification: Identifying bias-conflicting samples and amplifying their influence during target model training to prevent learning biased correlations.
The approach aims to balance the influence of bias-aligned and bias-conflicting samples, leading to an unbiased learning process. Extensive experiments demonstrated that Poisoner achieves superior debiasing performance compared to existing methods.

**Strengths:**

Innovative Use of Data Poisoning: The method leverages data poisoning, traditionally seen as a malicious tactic, for a beneficial purpose in debiasing.

Unsupervised Approach: Poisoner does not require labor-intensive bias feature annotations, making it more practical for real-world applications.

Iterative Refinement: The iterative nature of the method allows for continuous improvement in identifying and mitigating biases.

State-of-the-Art Performance: Extensive experiments show that Poisoner achieves superior performance in both fairness and accuracy on commonly used benchmarks.

Practicality: The method is designed to work without prior knowledge of bias types, making it adaptable to various datasets and scenarios.

**Limitations:**

I am not quite familiar with the field. However, based on the descriptions from the authors (the bias is caused by spurious features), it seems that many OOD generalization methods can be quite helpful (e.g., IRM), especially considering the cases where these methods may not required extra labels. I wonder if the authors could further discuss with the related methods in OOD generalization.

The suggested method seems to critically rely on the original biased model. If the biased model makes a lot of errors, the suggested method will not try to correct it. Instead, it will accumulate the errors and further bias the model. I think it is the key problem of the suggested method, and the authors should justify why the suggested method is robust to the errors in biased model.

It seems that the suggested method is computing demanding (two models, extra forward/backward propagation). Therefore, could the authors compare the required time of the suggested method and many other baselines.

**Suitability:**

2

---

### Meta-Review · Area_Chair_kdYP · 2024-07-08

**Recommendation:** Accept (Poster)
**Confidence:** 5

**Metareview:**

The paper introduces a novel debiasing method called "Poisoner," which leverages data poisoning to address biases in neural networks. This approach embeds biases into the training data to encourage models to learn and subsequently mitigate these biases. Reviewers praised the innovative use of data poisoning for a beneficial purpose, the unsupervised nature of the method, and the comprehensive experimental validation demonstrating state-of-the-art performance. However, they also noted several limitations, including the need for more comparisons with existing methods, potential issues with the complexity and computational demands of the approach, and some unclear methodological explanations. Despite these limitations, the reviewers generally found the work novel and impactful, leading to a mixed consensus with a lean towards acceptance.